# Taste-Active Peptides from Triple-Enzymatically Hydrolyzed Straw Mushroom Proteins Enhance Salty Taste: An Elucidation of Their Effect on the T1R1/T1R3 Taste Receptor via Molecular Docking

**DOI:** 10.3390/foods13070995

**Published:** 2024-03-25

**Authors:** Shiqing Song, Yunpeng Cheng, Jingyi Wangzhang, Min Sun, Tao Feng, Qian Liu, Lingyun Yao, Chi-Tang Ho, Chuang Yu

**Affiliations:** 1School of Perfume and Aroma Technology, Shanghai Institute of Technology, 100 Haiquan Road, Shanghai 201418, China; ssqing@sit.edu.cn (S.S.); 216072105@mail.sit.edu.cn (Y.C.); 216071164@mail.sit.edu.cn (J.W.); sunmin@sit.edu.cn (M.S.); fengtao@sit.edu.cn (T.F.); liu_qian_2021@163.com (Q.L.); lyyao@sit.edu.cn (L.Y.); 2Department of Food Science, Rutgers University, 65 Dudley Road, New Brunswick, NJ 08901, USA; ctho@ses.rutgers.edu

**Keywords:** straw mushroom, peptides from triple-enzymatically hydrolyzed straw mushroom proteins, molecular docking, UPLC-Q-TOF-MS/MS, T1R1/T1R3

## Abstract

The objective of our study was to analyze and identify enzymatic peptides from straw mushrooms that can enhance salty taste with the aim of developing saltiness enhancement peptides to reduce salt intake and promote dietary health. We isolated taste-related peptides from the straw mushroom extract using ultrafiltration and identified them using UPLC-Q-TOF-MS/MS. The study found that the ultrafiltration fraction (500–2000 Da) of straw mushroom peptides had a saltiness enhancement effect, as revealed via subsequent E-tongue and sensory analyses. The ultrafiltration fractions (500–2000 Da) were found to contain 220 peptides, which were identified through UPLC-Q-TOF-MS/MS analysis. The interaction of these peptides with the T1R1/T1R3 receptor was also assessed. The investigation highlighted the significant involvement of Asp223, Gln243, Leu232, Asp251, and Pro254 in binding peptides from triple-enzymatically hydrolyzed straw mushrooms to T1R1/T1R3. Based on the binding energy and active site analysis, three peptides were selected for synthesis: DFNALPFK (−9.2 kcal/mol), YNEDNGIVK (−8.8 kcal/mol), and VPGGQEIKDR (−8.9 kcal/mol). Importantly, 3.2 mmol of VPGGQEIKDR increased the saltiness level of a 0.05% NaCl solution to that of a 0.15% NaCl solution. Additionally, the addition of 0.8 mmol of YNEDNGIVK to a 0.05% NaCl solution resulted in the same level of saltiness as a 0.1% NaCl solution.

## 1. Introduction

Sodium chloride (NaCl) is recognized as a nutrient in the human diet and is a widely favored flavor enhancer while also providing an important source of the essential human nutrient sodium [1,2]. Despite its role in vital physiological processes, excessive dietary sodium intake is predominantly associated with elevated blood pressure and hypertension, elevating the risks of gastric cancer, cardiovascular disease, and chronic kidney disease [3,4]. Consequently, the imperative action of reducing dietary salt intake is evident. Currently, a healthier approach to salt reduction is the use of saltiness peptides or saltiness enhancement peptides, which are emerging salt reduction strategies in both theoretical research and practical application [5]. Saltiness enhancement peptides are characterized by their lack of inherent saltiness; however, in conjunction with sodium chloride, they significantly enhance the perception of saltiness in humans. This characteristic allows food products to maintain or even intensify their saltiness despite a reduction in sodium content when using saltiness enhancement peptides [6,7,8]. Furthermore, the composition of peptides, consisting of linked amino acids, enables them to supply the body with essential amino acids. Moreover, in comparison to other salts, saltiness enhancement peptides not only reduce irritation but also demonstrate a more pronounced saltiness enhancement effect [9].

In recent years, numerous studies have highlighted the saltiness enhancement properties of various food-derived peptides, such as saltiness enhancement peptides, umami peptides, Maillard-reacted peptides, and γ-glutamyl peptides [8]. Moore et al. discovered that various taste-modulating pyroglutamyl dipeptides extracted from mushrooms (e.g., pyroglutamylcysteine, pyroglutamylvaline, pyroglutamylaspartate, pyroglutamic acid, and pyroglutamylproline) exhibited notable saltiness enhancement effects at a concentration of 143 μmol/L in a mushroom broth model [10]. Yu et al. reported that grass carp skin collagen, subjected to enzymatic ultrafiltration, followed by a Maillard reaction with glucose, yielded Maillard-reacted peptides with saltiness-enhancing properties [11]. Additionally, EDEGEQPRPF, a taste peptide isolated from commercial plain fermented soybean curd, was found to enhance saltiness perception [6]. These findings underscore the feasibility of extracting saltiness enhancement peptides from both plant and animal sources, thereby providing a theoretical foundation for the extraction of such peptides from straw mushrooms.

To investigate the mechanism behind the enhancement of saltiness induced via peptides, it is essential to start with an understanding of how humans perceive saltiness. Salt taste perception is triggered by the presence of sodium and chloride ions in the oral cavity [12]. Prior investigations have revealed that human saltiness receptors include epithelial sodium channels (ENaCs) and transient receptor potential vanilloid 1 (TRPV1) [11,13]. The salt taste transduction pathway sensitive to amiloride is associated with ENaCs, whereas the amiloride-insensitive effect is mediated by TRPV1 [14,15]. Fu Yu et al. conducted molecular docking studies of collagen glycopeptides, produced through transglutaminase-induced glycosylation, with the saltiness receptor proteins ENaC and TRPV1 [5]. The study found that collagen glycopeptides can enhance saltiness perception. However, it should be noted that these findings may not be universally applicable, as some individuals have salt taste receptors that are insensitive to amiloride, resulting in a chloride-dominated and amiloride-insensitive salty response [12]. Additionally, it is noteworthy that the salt receptor TRPV1 responds not only to sodium ions (Na^+^) but also to potassium ions (K^+^) and ammonium ions (NH4^+^) [16]. Consequently, there are inherent limitations in exploring the mechanism of saltiness enhancement when using either ENaC or TRPV1 receptors for molecular docking with flavor-presenting peptides.

This study used umami taste receptors to screen taste-presenting peptides for their potential to enhance saltiness. The identified saltiness enhancement peptides were then validated through artificial sensory analysis. Extensive research has demonstrated the significant role of umami substances in saltiness enhancement, substantiating the effectiveness of molecular docking experiments on umami taste receptors (T1R1/T1R3). For instance, Xie et al. highlighted the impact of umami on enhancing saltiness at various concentrations [17]. Notably, ingredients such as hydrolyzed vegetable protein (HVP), yeast extracts, and specific amino acids like arginine, lysine, and taurine are recognized for increasing saltiness perception while concurrently reducing sodium chloride (NaCl) dependency [18]. Beyond these established flavor enhancers, umami peptides have been discerned as key contributors to saltiness enhancement. Xie et al. utilized molecular docking techniques to screen umami peptides from Ruditapes philippinarum and ham, identifying peptides such as KEMQKN, NGKET, RGEPNND, AHSVRFY, LSERYP, NRTF, TYLPVH, EV, AGAGTP, and GPAGAGPR for their ability to enhance saltiness [17]. Similarly, Shan et al. isolated taste peptides from yeast extracts and conducted molecular docking with the umami taste receptor proteins T1R1/T1R3, successfully identifying peptides that possess saltiness enhancement properties [18]. These findings highlight the important function of umami compounds in enhancing the perception of saltiness. Molecular docking with umami receptors is suggested as a potential technique for screening saltiness enhancement peptides.

The aim of this study was to determine the molecular weight range in which the peptides that enhance the saltiness of straw mushrooms are primarily found. We then analyzed and verified their saltiness enhancement effect by isolating and identifying peptides from the straw mushroom using sensory-guided analysis combined with UPLC-Q-TOF-MS/MS. The identified taste peptides were then molecularly docked with the T1R1/T1R3 umami receptors and their mechanism of action was analyzed. This study develops the potential for enhancing saltiness using straw mushroom peptides and provides a scientific basis for the further development of saltiness-enhancing peptide products.

## 2. Materials and Methods

### 2.1. Materials and Chemicals

Dried straw mushrooms were bought from Qingyuan High Mountain Agricultural Products Whole Sale Department. Food-grade salt (Deep Well Rock Salt) was purchased from China National Salt Industry Group Co., Ltd. (Beijing, China). Pectinase, cellulase, and papain were bought from Yuan Ye Biological Ltd. (Shanghai, China) Purified water was purchased from Watsons. Three peptides, DFNALPFX, YNEDNGIVK, and VPGGQEIKDR, were purchased from Hangzhou ALL PEPTIDE Biology Co., Ltd. (Hangzhou, China).

### 2.2. Extraction of Taste Peptides from Straw Mushrooms

The flavor peptide in straw mushrooms was extracted through step-by-step enzymatic hydrolysis. Firstly, the straw mushrooms were crushed and sieved. An appropriate amount of deionized water was added, and the two were mixed with a material–liquid ratio of 1:20. Then, the sample solution was bathed in water at 90 °C for 5 min and cooled to room temperature before use. We used HCl and NaOH to adjust the pH of the solution. In particular, we subtracted the perception of saltiness due to the NaCl introduced as a result of pH adjustment from all results. The pH was adjusted to 3.5, and pectinase 0.5% was added for enzymatic hydrolysis at 50 °C for 1 h. Next, the pH was adjusted to 4.5, and cellulase 0.5% was added for further enzymatic hydrolysis at 50 °C for 1 h. Subsequently, the pH was adjusted to 6.5, 0.4% papain was added, and enzymatic hydrolysis continued at 50 °C for 2 h. After that, the temperature was kept at 90 °C for 5 min to deactivate the enzyme and terminate the enzymatic reaction. Once the samples reached room temperature, the enzymatic hydrolysate was centrifuged at a speed rate of 7000 r/min for 15 min, and the supernatant was then stored for later use. The degree of hydrolysis (DH) of the supernatant after centrifugation was measured and combined with the solids’ content to determine the extent of enzymatic digestion.

The method chosen for determining the degree of hydrolysis was the formaldehyde titration method, which was calculated using the following formula:(1)$$DH(\%)=\frac{C\times \left(V_{1}-V_{2}\right)\times V/10}{m\times M\times 8}$$
where the following applies: *C* is the concentration of standard solutions of sodium hydroxide, mol/L;$\;V_{1}$ is the volume of sodium hydroxide standard solution consumed in the titration of the sample dilution to the end point by adding formaldehyde, mL;$\;V_{2}$ is the volume of sodium hydroxide standard solution consumed by adding formaldehyde titration to the end point in a blank test, mL; *m* is the weight of the sample, g; *M* is the protein content of the ingredients, g/100 g; and *V* is the total volume of enzymatic supernatant, mL.

### 2.3. Purification of the Peptides via Ultrafiltration

According to the reported method, the three samples were centrifuged at 4 °C, 7000 r/min, for 15 min to obtain the supernatant [11]. The supernatant was put into the ultrafiltration membrane separately. The samples were separated using an ultrafiltration system equipped with 500 Da and 2000 Da ultrafiltration membranes. Fractions with molecular weights of <500 Da, 500–2000 Da, and >2000 Da were collected.

### 2.4. Identification of the Taste Peptides

The freeze-dried portion of the ultrafiltration separation with the strongest saltiness enhancement effect was dissolved in deionized water. The fraction with the best saltiness enhancement effect was applied to UPLC-Q-TOF-MS/MS for peptide analysis. A sample of 30 mg was weighed, and 100 μL of 0.1% TFA was added and sonicated for 20 min; then, it was centrifuged at 17,000× *g* to obtain the supernatant for later use before the TIP was rinsed 10 times with 50 μL of 60% ACN/0.1% TFA. Then, the TIP was washed 10 times with 10 μL of 0.1% TFA. Next, the TIP was inhaled and drained 20 times. The liquid was drained, and the TIP washing continued with 10 μL of 0.1% TFA 5 times. Finally, the TIP was washed with 10 μL of 60% ACN, and 0.1% TFA eluted the peptide to a new EP tube, which was then vacuum-dried and tested on the machine using a preparative column (75 μm index, 150 mm, packed with Acclaim Pep Map RSLC C18(Thermo Fisher Scientific, Massachusetts, MA, USA), 2 μm, 100 Å, nano Viper). The above-treated peptides were dissolved with a 20 μL dissolved solution (0.1% formic acid, 5% acetonitrile), fully shaken, vortexed, and centrifuged at 13,500 rpm at 4 °C for 20 min. Then, the supernatant was transferred to the sample tube, and 8μL of pipette red was applied for mass spectrometry identification. Nitrogen was used as an atomizer and auxiliary gas. TOF-MS/MS (Agilent 6530, Santa Clara, CA, USA) was performed, and the mass range was *m*/*z* 100−1250 Da. PEAKS was used for data retrieval, and the amino acid sequences of the peptides were identified by matching them with the UniProt protein database. Data were analyzed based on the NCBI_Volvariella_validated.fasta database provided by Shanghai Research Dog Technology Co., Ltd. (Shanghai, China).

### 2.5. Molecular Docking (MD) of the Identified Peptides and T1R1/T1R3

The PDB Protein Library contains X-ray crystal diffraction and nuclear magnetic resonance (NMR) structural data uploaded by scientists for a number of currently known proteins or nucleic acids [19]. The amino acid sequences of the T1R1/T1R3 protein dimers were obtained from the UniProt website (T1R1, UniProtKB: Q7RTX1; T1R3, UniProtKB: Q7RTX0). The T1R1/T1R3 receptor model used in this study was obtained via homology modeling with the Modeller v9.19 program, using the crystal structure 5X2M as a template and optimizing the model for molecular dynamics with the Amber14 force field, as it was used by Song et al. [20]. Three-dimensional models of taste peptides identified by Q-Tof-MS/MS were built in PyMol 2.2.0, and these peptides were docked with the T1R1/T1R3 receptors in the program Autodockvina (The Scripps Research Institute Molecular Graphics Laboratory, La Jolla, CA, USA). Prior to molecular docking, we pre-treated the protein receptor and straw mushroom peptides’ ligand, including the dehydrogenation of the receptor ligand, the prediction of possible docking sites for the ligand, and the optimization of hydrogen bonding. The docking position at the T1R1/T1R3 site was x, y, z = 47.163, 35.635, 22.891. For each peptide docking with the T1R1/T1R3 protein receptor, 10 docking procedures were set up, and one peptide–protein conformation with the lowest binding energy was selected from the results, which was finally imported into PyMol for further analysis.

### 2.6. Sensory Assessors

The Sun et al. sensory method was used as a baseline, with some modifications based on it [21]. Sixteen sensory personnel were recruited from our laboratory to form a sensory panel, including 4 men and 12 women, aged 21 ± 2 years. None of them had oral diseases, their sense of taste was normal, and they were not usually alcohol or tobacco users. The training and evaluation of the sensory team were carried out according to the ISO 8586 standard [22]. The sensory panel rated the different concentrations of organoleptic references of Zhang et al. [23]. The following five flavor reference solutions were prepared to enable the sensory personnel to reach a consensus on the different taste intensities. The concentrations of the reference solutions were as follows: citric acid solution (0.430 mg/mL); sucrose solution (5.76 mg/mL); aqueous quinine sulfate solution (0.0325 mg/mL); sodium chloride solution (1.19 mg/mL); and sodium glutamate (0.595 mg/mL). To eliminate the effect of each sample’s odor, the samples had to be evaluated by the sensory personnel using nose clips in all sensory experiments.

### 2.7. Saltiness Was Compared Using the QDA Method

According to the Shan et al. method, we performed quantitative descriptive analysis (QDA) of the saltiness and umami flavors of different samples [18]. In order to distinguish the taste differences between different samples, we referred to the GB/T 12315-2008 standard [24]. The evaluator accepted three or more samples at the same time, and the order was randomized. Sample serial numbers were generated using a three-digit random code. After the panelists had tasted each sample, the samples were scored according to their perceived intensity of taste using a nine-point scale on which 1–3 was weak, 4–6 was moderate, and 7–9 was strong.

### 2.8. Comparing Relative Saltiness Using the Two-Alternative-Forced-Choice (2-AFC) Method

Control and test samples with the same salt concentration were evaluated using the 2-AFC design according to ISO-5495:2005 [25] with a risk of error of 5% [26]. This method was used to analyze the saltiness of three ultrafiltration samples. In subsequent experiments, the sensory team members were trained to familiarize themselves with assessing the salty intensity of the samples using the 2-AFC method. Referring to Sun et al., we set up six sets of salt solutions, D1–D6, with different concentrations as follows: D1, 2.03 g/L; D2, 2.99 g/L; D3, 4.39 g/L; D4, 6.45 g/L; D5, 9.48 g/L; and D6, 13.94 g/L [21]. The sensory personnel were required to evaluate one set of solutions at a time, and each set of solutions consisted of a salt solution of the same concentration (5 mL) and a solution to be tested (5 mL). After tasting, the sensory personnel had to choose the saltier sample as the result of the experiment. Since a two-choice test was employed, the sensory personnel evaluated the samples in the order of the gradient from the lowest to the highest salt concentration. All experiments were conducted at 25 ± 1 °C, and the sensory personnel were asked to rinse their mouths before evaluating the different groups of solutions to ensure the accuracy of the experiment.

### 2.9. Rating Saltiness Intensity Using the General Labeled Magnitude Scale (gLMS)

The sensory personnel were required to specifically evaluate the intensity of saltiness in different samples using the general labeled magnitude scale (gLMS) method, a psychophysical evaluation tool that requires sensory personnel to grade the intensity of perceived taste intensity along a vertical axis labeled with adjectives [27]. The descriptors it contains range from barely detectable = 1.38, weak = 5.75, moderate = 16.22, and strong = 33.11 to very strong = 50.12, and the strongest sensation of any type = 100 [28,29]. The experiment was conducted with only descriptors given to the sensory personnel, not numbers, and then the sensory personnel evaluated the results of the experiment based on the descriptors provided by the sensory evaluators. The descriptors provided by the sensory evaluators were then used to transform the experimental results into numbers and statistics, and the mean of the scores was used to indicate the intensity of the saltiness of the different samples. This method is used for the specific and quantitative sensory evaluation of the salinity of different ultrafiltration samples.

Reference was made to the method of Sun et al. with slight modifications [21]. For the formal experiments, the sensory personnel were required to evaluate one group of solutions at a time using the gLMS method, and each group consisted of a salt solution with different concentrations (D1–D6) and a solution with different molecular weights of taste peptides of straw mushrooms added. Finally, the results of the salt solutions of the same concentration were put together and analyzed. The sensory personnel put the samples (5 mL) in their mouths to revolve them for 10 s and then spit them out, and then they evaluated the intensity of the saltiness of the different samples using the gLMS method. In order to reduce experimental error, each set of experiments was repeated three times, and the final results were averaged.

Saltiness intensity was calculated according to the method of Sun et al. [21]. The natural logarithmic mean ratings of assessors’ saltiness intensities were calculated, and the power function between the salt concentration and saltiness intensity was obtained.
*lnI* = *lnk* + *nlnC*
where the following applies: *I* is the sensory intensity, in this case, the saltiness intensity of the salt solution; *C* is the stimulus level, in this case, the salt concentration; *n* is the power exponent; and *k* is a proportionality constant.

### 2.10. Evaluation of the Saltiness Enhancement Effect of Synthetic Peptides and Their Threshold Determination

In order to determine the saltiness enhancement effect of the synthetic peptides, we prepared a synthetic peptides solution by referring to the method of Zhang et al. with slight adjustments [23]. We dissolved 1 g of synthetic peptide in a 0.5% salt solution to make a 1 g/L salt solution containing the peptide. The salt solution containing the synthetic peptides was subjected to sensory evaluation using the QDA method described previously. Among the solutions, the 0.5% NaCl solution was assigned a score of 5 and served as a blank control group. In order to ensure the reasonableness of the results, we also measured the E-tongue of these solutions.

The threshold determination of synthetic peptides was determined via taste dilution (TD) analysis. In this test, we determined the taste thresholds of synthetic peptides in water and in a 0.5% NaCl solution, respectively. The peptide solution was first gradually diluted with deionized water at a ratio of 1:1 (*v*/*v*). Sensory evaluation was tested using triangulation, whereby the sensory personnel were required to taste these diluted peptide solutions continuously until it was not possible to distinguish the difference in taste between the sample solution and the two cups of deionized water. The number of dilutions at this point is the TD value, and the results were averaged across all sensory personnel evaluations [30].

### 2.11. Dose–Response Evaluation of the Saltiness Taste for Synthetic Peptides Solutions with NaCl Solutions

With reference to Song et al.’s approach, we made some modifications [20]. To verify the saltiness enhancement effect of the synthetic peptides, we prepared a series of three synthetic peptide solutions at different concentrations (0, 0.2, 0.4, 0.8, 1.6, and 3.2 mM). They were each prepared with a NaCl solution (0.5 g/L) at a 1:1 ratio to form a solution to be tested for the assessment of saltiness. The QDA method continued to be used, allowing sensory personnel to rate the saltiness of these samples on a scale of 1–9. In this case, scores of 1, 5, and 9 represent the saltiness enhancement intensity of 0.5 g/L, 1 g/L, and 1.5 g/L NaCl solutions, respectively.

### 2.12. E-Tongue Analysis

Minor modifications were made based on the method of Yan et al. [31]. The E-tongue (TS-5000Z, Intelligent Sensor Technology, Inc., Kanagawa, Japan) system from Japan was used, and the device was equipped with 6 sensors: AAE, CAO, CTO, COO, AE1, and GL1 for umami, sour, salty, bitter, astringent, and sweet, respectively. In order to make the experimental results more accurate, the whole testing process was carried out at an ambient temperature of about 25 °C. Before the start of the experiment, the E-tongue was debugged in the following sessions: activation, initialization, calibration, etc. The samples were configured with a reference solution of 30 mM of KCl and 0.3 mM of tartaric acid. At the beginning of the experiment, the liquid to be measured was poured into the special beaker for the E-tongue, and the liquid to be measured was placed alternately with the cleaning liquid on the E-tongue autosampler. The data collection of each sample was repeated 4 times, the collection time was 120 s, and the minimum error value of 3 times was taken as the measurement data of each sample.

### 2.13. Statistical Analysis

All experiments were repeated three times (*n* = 3). The results are expressed as means ± standard deviations. All data were statistically analyzed using SPSS Statistics 27.0.1 (SPSS Inc., Chicago, IL, USA), and differences between groups were analyzed via a one-way ANOVA using Duncan’s post hoc test (*p* < 0.05). Plotting was performed using Origin 2023 (Origin Lab, Northampton, MA, USA).

## 3. Results and Discussion

### 3.1. Solid Content and Degree of Hydrolysis of Straw Mushroom Peptides

Previous studies have mainly used single-enzyme or two-enzyme sequential methods to digest plant proteins. Luo et al. used a single-enzyme digestion method to extract chlorella proteins, while the efficient extraction of soy protein active peptides from soybeans was achieved using a double-enzyme digestion method [32,33]. In this study, we selected the optimal enzyme combination among single-enzyme enzymatic digestion, double-enzyme enzymatic digestion, and triple-enzyme enzymatic digestion based on the analysis of solid content, peptide molecular weight distribution, E-tongue results, and the salting effects of the straw mushroom’s enzymatic supernatant. The chosen optimal enzymatic digestion method was subsequently employed for the following experimental steps. Three enzymes, namely pectinase, cellulase, and papain, were selected for the experiments. The enzyme combinations for each group during enzymatic digestion were as follows: papain; cellulase and papain; and pectinase, cellulase, and papain.

Figure 1 illustrates the solids content and degree of hydrolysis in the supernatant after enzymatic digestion using various enzyme combinations. The solids content in the supernatant following single-enzyme digestion was 3.5%, which did not significantly differ from the solids content after double-enzyme digestion (3.6%). However, the solids content in the supernatant after triple-enzyme digestion reached 4.8%, signifying a notable difference compared to the other two groups and representing the highest solids content among the three enzymatic methods. In terms of the degree of hydrolysis, significant differences were observed among the three groups. The highest degree of hydrolysis was attained with triple-enzyme digestion at 50.1%, followed by double-enzyme digestion at 40.1%, with the lowest degree of hydrolysis recorded in single-enzyme digestion at 32.8%. These disparities may be attributed to the distinct properties of cellulase and pectinase. Pectin forms cross-links with cellulase, hemicellulose, lignin, and plant cell tissue proteins, resulting in strong bonding between adjacent fiber cells and inherent morphological traits [34]. As a result, the application of pectinase as a pretreatment on the raw material can partially degrade intercellular pectin, thereby easing fiber separation. On the other hand, cellulase is a highly efficient biocatalytic enzyme that degrades cellulase into reducing sugars [35,36]. Extensive research has demonstrated that the addition of cellulase can reduce lignocellulosic polymerization, increase pore space, and, when combined with papain, enhance the contact between papain and the substrate [35,37,38]. The concurrent use of pectinase and cellulase results in a more comprehensive degradation of the straw mushroom’s cell wall, enabling papain to more effectively degrade straw mushroom proteins and yield an increased quantity of straw mushroom polypeptides. Figure 1 unambiguously demonstrates that the solids content and degree of hydrolysis in the product supernatant were the highest following enzymatic hydrolysis using the triple-enzyme method, confirming its efficiency in enzymatic hydrolysis.

### 3.2. Free Amino Acid Content and Molecular Weight Distribution of Straw Mushroom Peptides

Amino acids serve not only as participants in human metabolism but also as pivotal flavor-contributing compounds, playing an integral role in the taste characteristics of food [39]. The contents of free amino acids in various flavor products following enzymatic digestion via distinct methods are tabulated in Table 1. Complete data concerning free amino acid content can be found in Appendix A. Among these compounds, free amino acids are particularly essential as active constituents in the umami flavor of edible mushrooms. Based on their flavor attributes, amino acids can be categorized into four classes: umami, sweet, bitter, and tasteless. The content within each class was quantified, with amino acids categorized as follows: umami (aspartic acid and glutamic acid), sweet (threonine, serine, glycine, alanine, and proline), bitter (valine, methionine, isoleucine, leucine, phenylalanine, histidine, and arginine), and tasteless (cysteine, tyrosine, and lysine) [40]. As illustrated in Table 1, triple-enzymatic digestion yielded a higher quantity of free amino acids compared to the other treatments. This can be attributed to the synergistic action of cellulase and pectinase, which enhanced the degradation of plant cell walls, thereby releasing a greater quantity of free amino acids [34].

Appendix A reveals the distinct molecular mass distribution of peptides within the preparations obtained through different enzymatic digestion methods. The notable increase in the fraction with a molecular mass of less than 500 Da following triple-enzymatic digestion can be attributed to the triple-enzymatic digestion conditions’ effectiveness in breaking down larger molecules into smaller ones. Notably, the molecular masses of amino acids typically fall below 500 Da. When considering this information alongside Appendix A, it becomes evident that the most substantial increase in amino acid content after triple-enzymatic digestion was observed when utilizing the three distinct extraction methods, consistent with the findings from the analysis of free amino acids. Moreover, the flavor of straw mushrooms is influenced by the presence of numerous flavor compounds [41]. A significant subset of these compounds comprises flavor-presenting peptides, which are oligopeptides with a molecular mass below 3000 Da [42]. As indicated in Appendix A, the fractions below 3000 Da are the most abundant following enzymatic digestion using three types of enzymes.

### 3.3. Evaluation of the Saltiness Enhancement Effect of the Enzyme Solution of Straw Mushroom

To examine the potential saltiness enhancement effects of straw mushroom enzymatic peptides, these peptides were added to a 0.5% NaCl solution for sensory and E-tongue evaluation. The outcomes of our experiments are presented in Figure 2. Specifically, Figure 2A,B display the results of E-tongue and sensory evaluations of saltiness and umami for the three enzymatic supernatants from straw mushrooms. The sensory evaluation employed the previously described nine-point scale, with a score of five signifying detectable saltiness (as indicated with the dotted line in Figure 2B). Figure 2A reveals that the three enzyme digests of straw mushrooms exhibited no perceivable saltiness and a decreasing umami intensity in the sequence of 15.81, 13.81, and 13.21, respectively. Figure 2B further illustrates that these enzyme digests were devoid of saltiness, according to sensory assessors, but still retained umami. Notably, saltiness enhancement peptides are a category of taste peptides that lack inherent saltiness but heighten the perception of saltiness [5]. Previous studies have suggested that the presence of umami substances might intensify the perception of saltiness in a saline solution [43]. To assess whether the extracted straw mushroom peptides possess saltiness enhancement properties, we introduced the enzymatically digested supernatants of straw mushrooms into a 0.5% salt solution for an investigation of their saltiness-enhancing effects.

Figure 2C,D depict the results of E-tongue and sensory evaluations conducted on the three enzymatic supernatants of straw mushrooms following their addition to a 0.5% NaCl solution. In Figure 2C, all three enzyme-digested supernatants of straw mushrooms exhibited saltiness enhancement in the 0.5% NaCl solution environment compared to the saltiness intensity of the 0.5% NaCl solution (denoted as 4.58 with the dotted line). This outcome aligns with the findings of Shan et al. [18]. Among these, the triple-enzyme digestion demonstrated the most pronounced saltiness enhancement, elevating the saltiness value of the 0.5% NaCl solution from 4.58 to 6.49, consistent with sensory evaluation results. Figure 2D further indicates that sensory assessors considered the peptides from the triple-enzymatic hydrolysis of straw mushrooms more effective in enhancing the saltiness value of 0.5% NaCl. It is noteworthy that, in a 0.5% NaCl solution environment, the umami of the enzyme solution decreased, while the saltiness was heightened.

In light of these experimental results, it becomes evident that triple-enzyme enzymatic hydrolysis, characterized by a higher degree of hydrolysis, can yield more flavor-presenting peptides and offers superior saltiness enhancement effects. This suggests that triple-enzyme digestion could be a promising method for preparing saltiness enhancement peptides. To delve deeper into the saltiness enhancement potential of straw mushroom enzyme peptides, we performed ultrafiltration on the triple-enzyme-digested supernatants using 500 Da and 2000 Da ultrafiltration membranes, resulting in three fractions: A1 (<500 Da), A2 (500–2000 Da), and A3 (>2000 Da). These fractions were subsequently freeze-dried and prepared for further analysis.

### 3.4. Comparison of Saltiness Intensity Using the 2-AFC

The data were modeled under a binomial distribution, with a minimum of 12 agreeing judgments required for significance (one-tailed binomial distribution, *n* = 32, α = 0.05). As depicted in Figure 3, more than 12 evaluators deemed samples with these fractions saltier than control samples at all salt concentrations (D1–D6). This indicates that all three ultrafiltration fractions can enhance saltiness in the salt solution. Notably, most panelists, across various salt concentrations, found samples with these fractions to be saltier than the control sample. As the saltiness enhancement did not significantly differ among the fractions, an in-depth analysis of gLMS scores was warranted [21].

### 3.5. Saltiness Intensity Evaluation Using the gLMS

The gLMS method is a psychophysical evaluation tool in which only descriptors, not numbers, are provided to the sensory evaluator during the experiment. The experimenter then performs numerical transformations and statistics on the results of the experiment, based on the descriptors provided by the sensory evaluator, and the degree of saltiness is expressed as the mean of the scores [27]. For gLMS saltiness intensity evaluation, the saltiness enhancement ability of these three ultrafiltration components was further investigated under the guidance of 2-AFC results. A repeated-measures ANOVA was employed to assess the significance of saltiness intensity ratings. Figure 4A illustrates fraction A2 significantly heightened saltiness intensity was observed across all salt concentrations compared to the other two fractions.

A1, on the other hand, substantially enhanced saltiness in the D2, D3, and D6 salt solutions but exhibited no significant enhancement for D5. Conversely, A3 displayed no apparent saltiness enhancement in all six salt concentrations, and A2 had no discernible enhancement for D4. In summary, the enhancement effects of different ultrafiltration fractions at various salt concentrations on saltiness intensity varied, with A2 demonstrating significant saltiness enhancement across concentrations from D1 to D6 [18].

Analyzing the gLMS results tentatively suggests that A2 is the most effective of the three ultrafiltration fractions in enhancing saltiness. Subsequently, saltiness intensity scores were logarithmically transformed to establish power functions correlating with salt concentration. The power functions, reflecting the relationship between the natural logarithm of salt concentration and the natural logarithm of saltiness intensity, are presented in Appendix A to help determine iso-saltiness concentrations among the fractions. The results reveal exponents ranging from 0.9207 to 1.0736, both less than the control. Consistent with Sun et al.’s findings, the ln–ln curves support the notion that all three ultrafiltration fractions enhance saltiness at varying salt concentrations [43]. These curves demonstrate a more pronounced increase in the lower salt concentration range, while achieving a higher salt reduction percentage becomes challenging at higher concentrations. In salt solutions of the D2, D3, and D4 concentrations, A2 exhibited the most prominent saltiness enhancement effect compared to the other fractions. Therefore, it was preliminarily concluded that A2 (500–2000 Da) is particularly good at enhancing saltiness.

### 3.6. Identification and Molecular Docking of Peptides

Through a comprehensive analysis of the preceding experiments, it was evident that the A2 fraction (500–2000 Da) from enzymatic supernatants of straw mushrooms holds the most prominent saltiness enhancement effect. It is clear from this study that peptides derived from the triple-enzymatic hydrolysis of straw mushrooms impart umami, rather than saltiness. However, when introduced to a salt solution, they exhibit a saltiness-enhancing effect. This aligns with Xie et al.’s findings that substances with umami attributes can heighten saltiness in salt solutions [17]. To delve deeper into the saltiness enhancement effect of peptides from triple-enzymatic hydrolyzed straw mushrooms, we subjected the A2 fraction to UPLC-Q-TOF-MS/MS analysis to acquire specific peptide information [17]. Building upon the research of Shan et al., which revealed that peptides possessing inherent umami characteristics but lacking saltiness could be initially selected through docking with umami receptors T1R1/T1R3, our study delved deeper into the saltiness enhancement potential of these screened peptides [18]. The identified peptides were then subjected to molecular docking experiments to reveal the underlying molecular mechanisms. To investigate whether peptides derived from the triple-enzymatic hydrolysis of straw mushrooms possessed saltiness enhancement attributes, we pursued the identification of peptides within the A2 fraction and their molecular docking with T1R1/T1R3 [44].

Peptide identification was performed via UPLC-Q-TOF-MS/MS. In this context, electrospray ionization in a high-resolution mass spectrometer was employed to dissociate peptides in the A2 fraction samples, generating fragment ions. Amino acid sequences were determined by aligning these fragments with straw mushroom protein libraries on UniProt [45]. The identified peptides, with molecular weights of <2 kDa and containing 5–10 amino acid residues, totaled 220. Subsequently, molecular docking simulations with T1R1/T1R3 were carried out to screen the peptides. Peptides with matching mass-to-charge ratios were initially screened based on their compatibility with b and y ions. Subsequently, eight potential taste peptides were selected based on molecular docking scores: DFNALPFK (DF8; mass: 875.04; *m*/*z*: 574.25), VPGGQEIKDR (VP10; mass: 1097.5829; *m*/*z*: 366.87), GVGPFDDDR (GV9; mass: 976.425; *m*/*z*: 489.22), SEHEENGYAV (SE10; mass: 1133.46; *m*/*z*: 567.75), YNEDNGIVK (YN9; mass: 1050.50; *m*/*z*: 526.26), IDNEPEFRWA (ID10; mass: 1275.59; *m*/*z*: 638.81), DKLHEGIK (DK8; mass: 938.52; *m*/*z*: 313.85), and IGDEAAENRN (IG10; mass: 1254.42; *m*/*z*: 544.75). Their MS/MS data are illustrated in Appendix A.

The use of molecular docking with taste peptides, employing T1R1/T1R3 receptors, offers an avenue to explore the taste properties of these peptides and introduces novel perspectives for taste peptide research [46]. The two-dimensional diagrams of the eight peptides are displayed in Figure 5, and the molecular docking scores are provided in Table 2. These scores demonstrate that all eight peptides can bind to the Venus flytrap (VFT) binding domains of T1R1/T1R3 [47]. Throughout the docking process, the T1R1/T1R3 receptor structure remains fixed, while the peptide structure adapts its conformation. The docking process allows for diverse conformational changes in the peptides, with the binding model featuring the lowest binding energy (highest score) being selected [48]. The docking energies are ranked as follows: DF8 (−9.2 kcal/mol), VP10 (−8.9 kcal/mol), YN9 (−8.8 kcal/mol), GV9 (−8.7 kcal/mol), SE10 (−8.7 kcal/mol), ID10 (−8.6 kcal/mol), DK8 (−8.6 kcal/mol), and IG10 (−8.3 kcal/mol). As demonstrated by Liang et al., lower binding energies in molecular docking results indicate more stable conformations [44]. Notably, DF8 exhibits the lowest docking energy (highest score) with T1R1/T1R3. The molecular docking 2D diagrams illustrate that peptides from triple-enzymatic hydrolyzed straw mushrooms chiefly engage in hydrogen bonding and hydrophobic interactions when binding to the T1R1/T1R3 receptor. However, the binding characteristics of different peptides to the T1R1/T1R3 receptor are not uniform, suggesting that taste peptide properties cannot be exclusively determined based on binding energy [18]. Apart from determining the most stable binding conformation through binding energy, molecular docking studies offer insights into the peptide–receptor binding site [49,50]. A total of 45 amino acid residues in T1R1/T1R3 play pivotal roles in interactions with peptides from triple-enzymatic hydrolyzed straw mushrooms. As depicted in Figure 6, the actively docked residues predominantly include Gln 227, Lys 215, Gln 223, Asn 223, Gln 243, Glu 226, Ser 167, Val 168, Glu 151, Lys 171, Lys 222, Glu 219, Tyr 242, Asn 262, Lys 176, and Ala 255. Notably, the amino acids Asp, Glu, Ser, and His make the most substantial contributions to molecular interactions. In conclusion, the active sites highlighted in this molecular docking study serve as a foundational step for the preliminary screening and prediction of unknown taste peptides in subsequent research. As supported in the existing literature, protein receptors and peptide ligands in molecular docking establish connections through hydrogen bonding and peptide ligands, in addition to hydrogen bonding, and they also engage in van der Waals interactions, electrostatic interactions, alkyl interactions, and hydrogen interactions [51,52]. Consistent with our findings, the binding forces between the peptides from triple-enzymatic hydrolyzed straw mushrooms in this study and the protein receptor’s active residues can be observed in 2D diagrams, comprising primarily carbon–hydrogen bonding, conventional hydrogen bonding, and charge attractions, followed by salt bridges, π-alkyl groups, π-anions, and alkyl groups.

### 3.7. Effect of Synthetic Peptides on Saltiness in Salt Solutions

The length of the peptide chain also determines the properties of the flavor-presenting peptide [53]. Among the ten peptides mentioned above, we selected the three peptides with the lowest molecular docking binding energies and strong interactions with receptor proteins from 8, 9, and 10 peptides, respectively, for synthesis: DF8, YN9, and VP10. To investigate whether the peptide has a saltiness enhancement effect, the extracted peptide was added to a salt solution of the same concentration, and the peptide–salt solution system was used to assess the saltiness enhancement effect of peptides [54]. To ascertain whether these synthesized peptides possess saltiness enhancement properties, they were introduced into a 0.5% salt solution for sensory and E-tongue evaluations. The results, presented in Figure 7, demonstrate that all three peptides can indeed enhance saltiness in the solution. Among them, VP10 exhibited the most potent effect, followed by DF8 and YN9. This observation aligns with the E-tongue results, substantiating that VP10 wields the most substantial saltiness enhancement effect. Furthermore, a notable correlation exists between the E-tongue outcomes and sensory evaluations, underscoring the utility of E-tongue in assessing peptide-induced saltiness enhancements [55].

The taste threshold of synthetic peptides was determined experimentally using the method mentioned earlier. The taste characteristics and threshold data for the synthetic peptides are outlined in Table 3. We examined the taste thresholds for these peptides in both pure water and a 0.5% NaCl solution. Beyond the mere realm of taste, peptides may contribute to a broader flavor profile [56]. Evidently, sour and astringent notes could be perceived in the identified peptides. This sourness may originate from residual free amino acids and acetate remnants during peptide synthesis [20]. Research indicates that umami amino acids, such as aspartic acid (Asp), can evoke sourness or umami if found in a free state [23]. It is worth noting that DFNALPFK and VPGGQEIKDR were characterized by umami or a faint umami taste. Notably, in pure water, DFNALPFK displayed the lowest threshold at 0.38 mM, consistently mirroring the molecular docking scores. Higher scores corresponded to lower thresholds, while lower scores indicate comparatively higher thresholds. In both pure water and the 0.5% NaCl solution, all three synthetic peptides demonstrated lower thresholds in the saline environment, affirming their saltiness enhancement effects. Among these, DFNALPFK exhibited the lowest threshold in the 0.5% NaCl solution, signifying its most pronounced saltiness enhancement effect. Notably, VPGGQEIKDR displayed saltiness characteristics and is likely a peptide with saltiness attributes.

### 3.8. Dose–Feedback Saltiness Enhancement Effects of Synthetic Peptides

The figure clearly displays significant variations in saltiness intensity as the concentration of synthetic peptides increases. Notably, within the concentration range of 0.4–0.8 mmol, there is a sharp rise in saltiness intensity in the NaCl solution. However, when the concentration of synthetic peptides surpasses a critical threshold (0.8–1.6 mmol), the effectiveness of saltiness enhancement begins to decline. From Table 3, it can be seen that the synthetic peptide has an acidic flavor, and apart from the fact that the peptide itself contains acidic amino acid residues, this may be related to residual chemical reagents such as sodium acetate introduced during the peptide synthesis process [20,57]. Consequently, sensory assessors may not perceive the changing saltiness. Figure 8 also reveals that, in the 0.4–0.8 mmol concentration range, both DF8 and VP10 exert a significant saltiness enhancement effect on the NaCl solution. However, as the concentration reaches the critical range of 0.8–1.6 mmol, the saltiness enhancement effect of DF8 and YN9 starts to decline, while VP10 continues to augment the saltiness intensity of the solution. Taken together, the saltiness enhancement effect of the three synthetic peptides increased with increasing concentration. Once the concentration of these peptides exceeded a certain level, their acidity became so strong that it masked the saltiness. This may be because the peptides themselves have an acidic taste, and if the peptides are concentrated too highly, their acidity masks the saltiness, giving the solution an overall sour taste [20]. In summary, this study underscores that peptides derived from the triple-enzymatic hydrolysis of straw mushrooms can indeed augment the saltiness of salt solutions. However, the quantity of added peptides should be carefully managed to achieve the optimal saltiness enhancement effect [5]. Notably, adding 0.8 mmol of YN9 results in a 0.05% NaCl solution achieving the saltiness intensity of a 0.1% NaCl solution. Furthermore, incorporating 3.2 mmol of VP10 elevates a 0.05% salt solution to the saltiness level of a 0.15% salt solution. This study’s results align with Xie et al.’s findings, in which peptides from ham enhanced a 0.3% salt solution to the saltiness of a 0.4–0.6% salt solution [17]. Similarly, a peptide extracted from curd (EDGEQPRPF) at a concentration of 0.4 mg/mL intensified a 50 mM NaCl solution to the saltiness of a 63 mM NaCl solution [6].

According to the results of the dose–feedback experiment, it was observed that DF8 (DFNALPFK) and VP10 (VPGGQEIKDR) exhibit superior saltiness enhancement effects. These effects were compared to those of previously reported saltiness-enhancing peptides such as KEMQKN, NGKET, RGEPNND, AHSVRFY, LSERYP, NRTF, TYLPVH, AGAGTP extracted from *Ruditapes philippinarum* and ham, GPAGAGPR [6,17], TPPKID, PKESEKPN, TEDWGR LPLQDAH, NEFGYSNR, and LPLQD obtained by Zhang et al. from chicken [58], SGCVNEL, EPLCNQ, and ESCAPQL isolated from *Stropharia rugosoannulata* and found by Chen et al. to have saltiness enhancement effects [59], and EV, AM, AVDNIPVGPN, and VDNIPVGPN in yeast extract, discovered by Liang et al. to enhance saltiness [60]. It was noted that these saltiness enhancement peptides share similar amino acid residues in their composition, including Asp(D), Lys(K), Pro(P), Glu(E), Phe(F), Gly(G), and Asn(N). Notably, Glu(E) and Asp(D) are umami amino acids, while Lys(K), Gly(G), Asn(N), and Phe(F) belong to the category of hydrophilic amino acid residues. Based on this preliminary analysis, we can speculate that peptides containing one or more umami amino acids or hydrophilic amino acid residues are highly likely to possess salt-enhancing properties.

## 4. Conclusions

In this study, we conducted a comprehensive investigation into the flavor peptides extracted from straw mushrooms using a three-enzyme enzymatic method. This research not only focused on the extraction process but also delved into the mechanisms through which these peptides contribute to flavor enhancement through molecular docking techniques. Ultimately, based on the outcomes of molecular docking, we identified and synthesized three peptides to examine their saltiness-enhancing properties. The findings revealed that peptides obtained from the triple-enzymatic hydrolysis of straw mushrooms exhibited umami qualities but lacked saltiness; however, when introduced to a 0.5% NaCl solution, these peptides displayed reduced umami and increased saltiness. This observation led us to speculate that peptides derived from triple-enzymatically hydrolyzed straw mushrooms might possess attributes for enhancing saltiness. To further understand the underlying molecular mechanisms, we assessed the potential of different molecular weight ranges in straw mushroom enzymatic solutions for enhancing saltiness across diverse concentrations of salt solutions. The results indicated that straw mushroom enzymatic peptides within the 500–2000 Da range had significant effects in enhancing saltiness. Based on the molecular docking binding energy score, eight peptides with saltiness enhancement potential were selected. Among these candidates, DFNALPFK, YNEDNGIVK, and VPGGQEIKDR emerged as the top contenders based on the analysis of active sites and amino acid residues involved in molecular docking interactions. Subsequently, we synthesized these three peptides to validate their ability to enhance saltiness; all three demonstrated positive effects, with VPGGQEIKDR exhibiting the most potent impact specifically at a concentration of 3.2 mM, which significantly elevated a 0.05% NaCl solution’s intensity.

## Figures and Tables

**Figure 1 foods-13-00995-f001:**
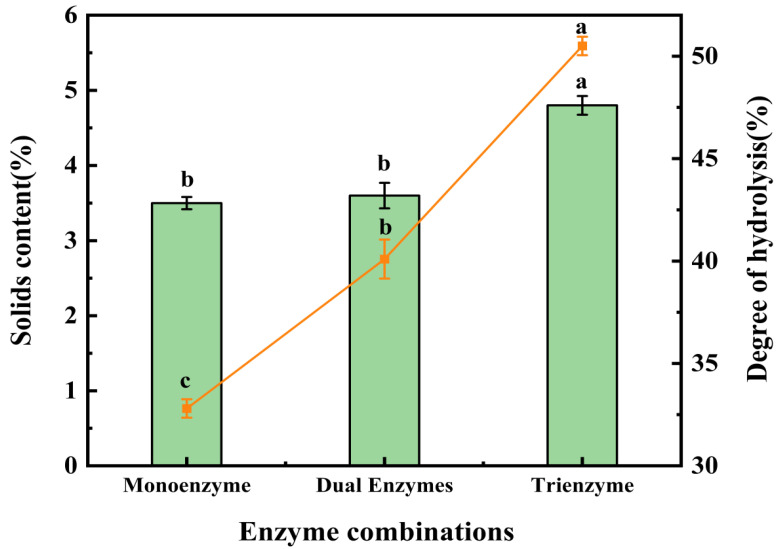
Results of solids content and the degree of hydrolysis (DH) after treatment with three enzymatic methods. The line matches the left y-axis (solids content), and the bar matches the right y-axis (DH). According to a one-way ANOVA, the same letter of the marker indicates that there is no statistical difference between the amounts of the solids content and DH in the supernatant obtained using different enzymatic methods (*n* = 3, *p* < 0.05).

**Figure 2 foods-13-00995-f002:**
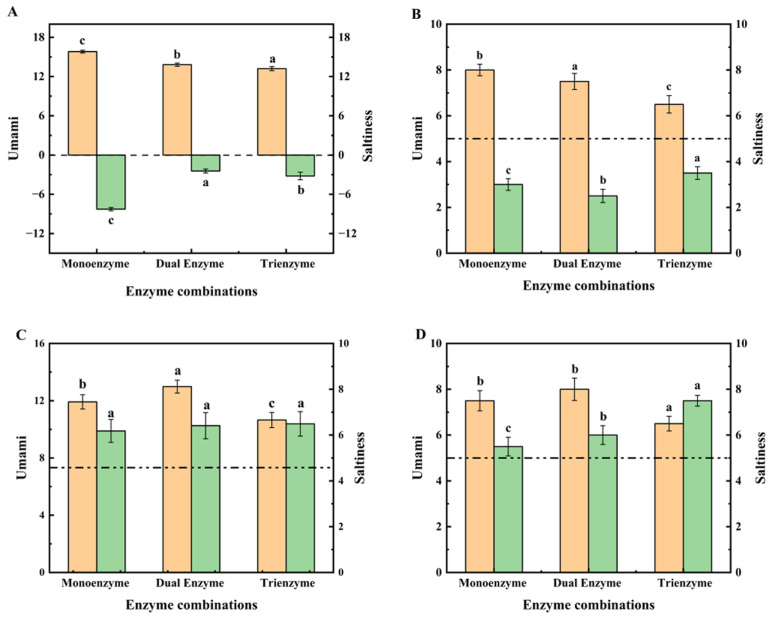
(**A**,**C**) are the results of the E-tongue evaluation, and (**B**,**D**) are the results of the sensory evaluation. The substrate in (**A,B**) is the enzyme supernatant, and the substrate in (**C**,**D**) is a mixture of a 0.5% NaCl solution and the enzyme supernatant. The dotted line in (**B**) indicates the score for the ability to taste saltiness. The dotted line in both (**C**,**D**) indicates the score for the 0.5% NaCl solution. According to a one-way ANOVA, the same letter of the marker indicates that there is no statistical difference between the amounts of the E-tongue value and sensory score in the supernatant obtained using different enzymatic methods. (*n* = 3, *p* < 0.05).

**Figure 3 foods-13-00995-f003:**
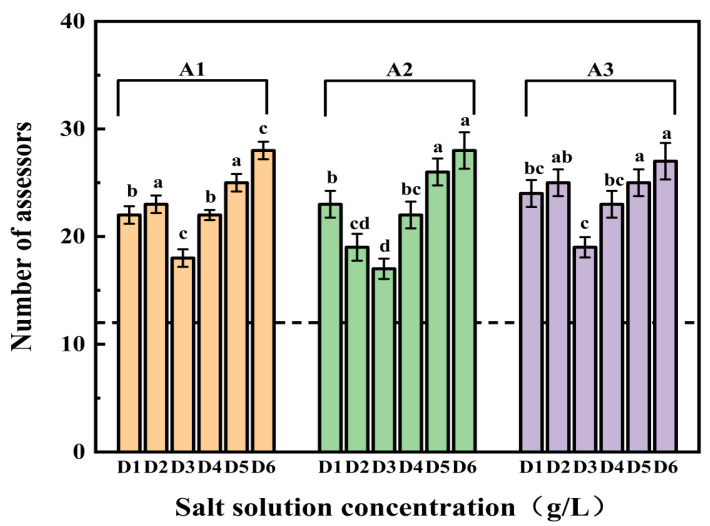
Results of the 2-AFC experiment. The vertical axis indicates the number of assessors who agreed that the different ultrafiltration fractions were saltier. The dashed line indicates the minimum number of agreement judgments required to identify the saltier samples in a comparison test at the α = 0.05 level. According to a one-way ANOVA, the marks with the same letter indicate that there is no statistical difference between the number of assessors at different concentrations of the salt solution (*n* = 3, *p* < 0.05).

**Figure 4 foods-13-00995-f004:**
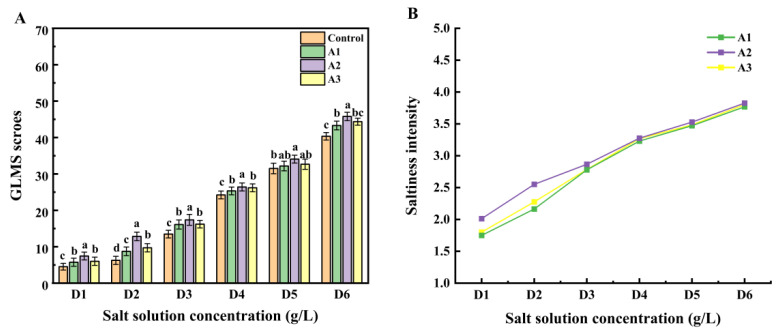
The gLMS scores (**A**) and the change of saltiness intensity (**B**) under three ultrafiltration fractions. According to a one-way ANOVA, the same letter of the marker indicates that there is no statistical difference between the amounts of the gLMS scores in different ultrafiltration fractions (*n* = 3, *p* < 0.05).

**Figure 5 foods-13-00995-f005:**
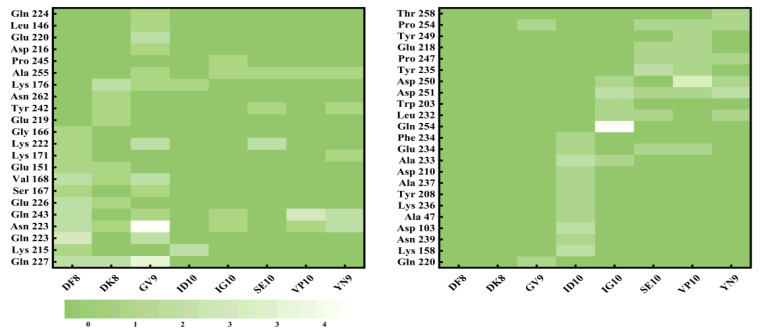
Statistics of the active sites of eight taste peptides interacting with the T1R1/T1R3 umami receptor.

**Figure 6 foods-13-00995-f006:**
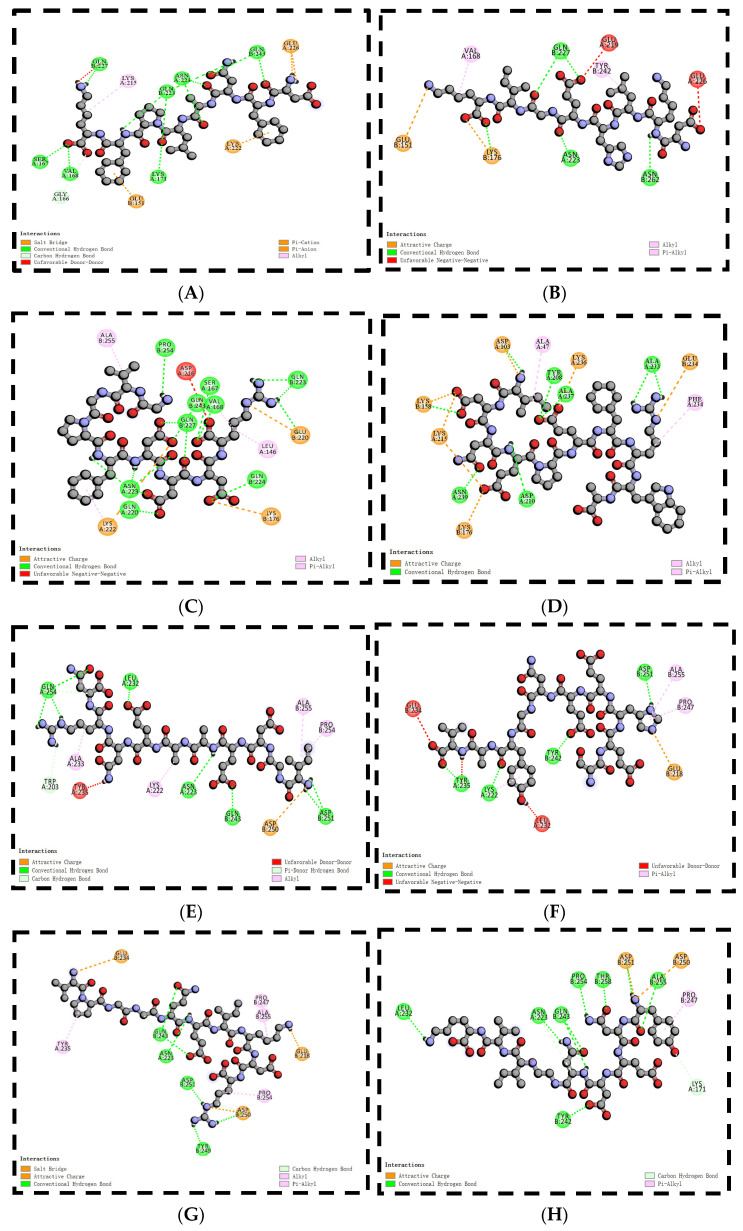
Two-dimensional diagram of the docking of DFNALPFK (**A**), DKLHEGIK (**B**), GVGPFDDDR (**C**), IDNEPEFRWA (**D**), IGDEAAENRV (**E**), SEHEENGYAV (**F**), VPGGQEIKDR (**G**), and YNEDNGIVK (**H**) with T1R1/T1R3.

**Figure 7 foods-13-00995-f007:**
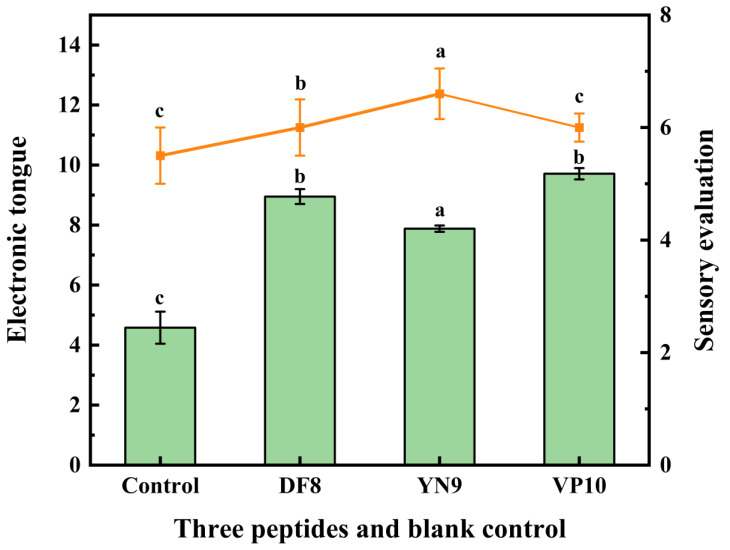
The saltiness enhancement characteristics of synthetic peptides. The bar matches the left y-axis (E-tongue), and the line matches the right y-axis (sensory evaluation). According to a one-way ANOVA, the same letter of the marker indicates that there is no statistical difference between the amounts of the E-tongue and sensory evaluation in different synthetic peptides (*n* = 3, *p* < 0.05).

**Figure 8 foods-13-00995-f008:**
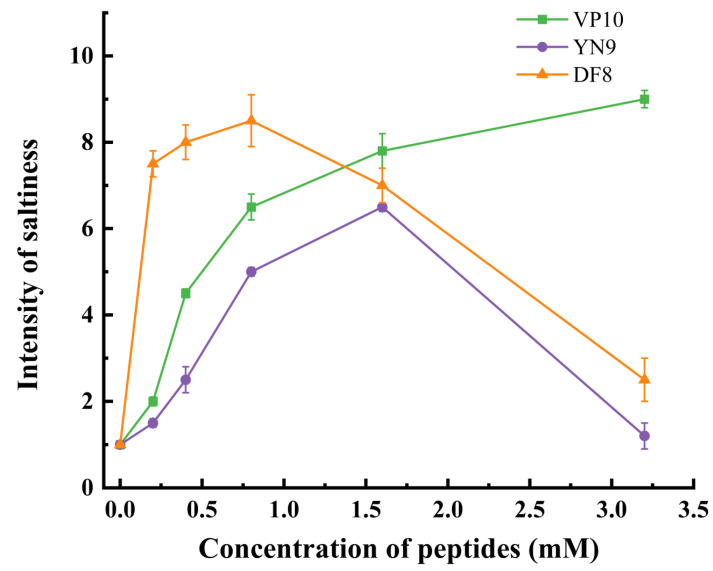
Dose–feedback results for three synthetic peptides. The green line represents VP10, the blue line represents YN9, and the orange line represents DF8. Using a one-way ANOVA (*n* = 3, *p* < 0.05).

**Table 1 foods-13-00995-t001:** Results of the free amino acid content in the supernatant.

Amino Acids	Content (mg/mL)		
	Monoenzyme	Dual Enzyme	Trienzyme
Bitter	1.13 ± 0.01 ^a^	0.88 ± 0.01 ^b^	1.06 ± 0.02 ^c^
Sweet	0.52 ± 0.01 ^b^	0.51 ± 0.01 ^b^	0.61 ± 0.02 ^a^
Umami	0.52 ± 0.02 ^b^	0.53 ± 0.02 ^b^	0.65 ± 0.01 ^a^
Tasteless	0.38 ± 0.01 ^b^	0.34 ± 0.01 ^b^	0.41 ± 0.02 ^a^
Total	2.55 ± 0.01 ^c^	2.26 ± 0.01 ^b^	2.74 ± 0.02 ^a^

According to a one-way ANOVA, the same letter of the marker indicates that there is no statistical difference between the amounts of the free amino acid content in the supernatant obtained using different enzymatic methods (*n* = 3, *p* < 0.05).

**Table 2 foods-13-00995-t002:** Results of the peptide molecular weight distribution.

	Peptide	Length	Affinity (kcal/mol)
1	DFNALPFK	8	−9.2
2	VPGGQEIKDR	10	−8.9
3	YNEDNGIVK	9	−8.8
4	IDNEPEFRWA	10	−8.6
5	GVGPFDDDR	9	−8.7
6	SEHEENGYAV	10	−8.7
7	DKLHEGIK	8	−8.6
8	IGDEAAENRN	10	−8.3

**Table 3 foods-13-00995-t003:** Taste attributes and threshold values of synthetic peptide.

Peptide	Taste Attribute	Threshold Value (mM)	
		Water	0.5%NaCl
DFNALPFK	Sore, astringent, umami	0.38 ± 0.03 ^b^	0.22 ± 0.02 ^c^
YNEDNGIVK	Sore, astringent	0.45 ± 0.02 ^b^	0.33 ± 0.05 ^b^
VPGGQEIKDR	Sore, astringent, weakly umami	0.56 ± 0.04 ^a^	0.41 ± 0.03 ^a^

Note: The values within the same column followed by different superscript letters are significantly different (*n* = 3, *p* < 0.05).

## Data Availability

The original contributions presented in this study are included in the article/Appendix A. Further inquiries can be directed to the corresponding author.

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
