# Peer review of "Taste-Active Peptides from Triple-Enzymatically Hydrolyzed Straw Mushroom Proteins Enhance Salty Taste: An Elucidation of Their Effect on the T1R1/T1R3 Taste Receptor via Molecular Docking"

_foods, 2024, doi:10.3390/foods13070995_

Round 1

Reviewer 1 Report

Comments and Suggestions for Authors

Comments for Authors

Here are the main two comments regarding the paper:

1.     The authors have discovered umami peptides from hydrolyzed straw mushroom proteins and demonstrated their saltiness enhancement effect. The approach of analyzing the umami and saltiness enhancement effects of hydrolyzed straw mushroom proteins through multiple sensory evaluation tests is commendable. However, there is a lack of discussion on whether the identified peptides genuinely contribute to the saltiness enhancement of hydrolyzed straw mushroom proteins (and if they are present in concentrations that contribute), and how much of the contribution can be explained by the three peptides. The molecular docking energies of ten peptides to T1R1/T1R3 do not significantly vary, making the selection of three peptides not strongly justified. Should not the selection of peptides for sensory evaluation consider both the docking data and the content of each peptide?

2.     This study raises ethical concerns that need to be explicitly addressed before accepting this paper from an ethical standpoint. Are the hydrolyzed straw mushroom proteins and the three peptide compounds used as sensory evaluation samples prepared for human consumption? There is no mention of ethical review for the sensory evaluation. The authors discuss that "this may be related to residual chemical reagents such as sodium acetate introduced during the peptide synthesis process" (Line 645). Was the use of synthetic reagents with potential residual risks ethically approved for human trials?

Minor comments include:

1.     Line 37: What does "T Sodium Chloride" refer to?

2.     Line 43: Aren't alternative salts, such as KCl, the most common current approach for salt reduction? The authors' statement that peptides are the most effective method feels out of touch with the current salt reduction market, "Currently, the most effective method for salt reduction involves the utilization of saltiness peptides, or saltiness enhancement peptides, which is an urgently needed salt reduction strategy in both theoretical research and practical applications."

3.     Line 116: What does "T Dried straw mushrooms" mean?

4.  Line 207: What is "GB/T 12315-2008"? Is it a standard taste solution kit? Please elaborate.

5.     Line 232: Why are the numbers in the descriptors not round numbers like 3, 5, 15, but rather specific like 1.38?

6.     Line 262: "1g" cannot start a sentence in Arabic numerals. Throughout the text, there are instances of awkward verb choices. A thorough English revision is necessary before acceptance. Consider using tools like Chat GPT for English corrections.

7.     Fig 2: The explanation of the differences between A, B, C, D in the figure legend is unclear. Please also explain what the dotted lines represent in the figure legend. Revise the figure legend to be concise and clear so that the results are understandable without reading the entire manuscript.

8.     Fig 3: The meaning of the vertical axis is unclear. The interpretation of the graph is not understandable.

9.     Are the umami and saltiness enhancement effects of hydrolyzed straw mushroom proteins truly due to taste? What about the impact of smell?

10.  Compared to previously reported saltiness enhancement peptides, are the amino acid sequences significantly different or similar? How does the strength of the saltiness enhancement effect compare? Please discuss.

Comments on the Quality of English Language

Throughout the text, there are instances of awkward verb choices. A thorough English revision is necessary before acceptance.

Reviewer 2 Report

Comments and Suggestions for Authors

This work reports on the use of mushroom extracted peptides to improve the salt sensation without the addition of a NaCl.

The work is well presented and relevant, however I believe that some questions should be addressed.

In material and methods - When extracting the peptides, the authors state that they use dH2O solution which makes me think that, before the hydrolysis, the pH of the solution is close to 7. Then the pH is adjusted to 3.5, 4.5, and 6.5. For this I am guessing (but should be described) that the adjustment is done with HCl, followed by NaOH?! my question is, how do you know that the saltiness increase is due to the peptides and not to the addition of NaCl by the interactions  between acid and base at the pH adjustment phase?

line 126 states "Then, the enzyme was incubated at 90 ℃ for 5 mins and allowed to cool to room temperature." What enzyme? and what is the reasoning of this step?

please use passive voice throughout, for example in line 213 change "we used this method..." to " this method was used...". Check throughout.

What is the brand and making of the e-tongue used? (add it in line 288 after the model)

Add in your graphs labels that the results are an average on n=16 or 3 or what ever the n is for each test +-std dev/std error (not sure what did you use) and that different subscript letter correspond to significant difference with p<0.05?! Some figures and tables have this info, some don't.

I am looking forward to seeing the improved manuscript.

Regards
